# Enhanced Switching Reliability of Sol–Gel-Processed Y_2_O_3_ RRAM Devices Based on Y_2_O_3_ Surface Roughness-Induced Local Electric Field

**DOI:** 10.3390/ma15051943

**Published:** 2022-03-05

**Authors:** Do-Won Kim, Hyeon-Joong Kim, Won-Yong Lee, Kyoungdu Kim, Sin-Hyung Lee, Jin-Hyuk Bae, In-Man Kang, Kwangeun Kim, Jaewon Jang

**Affiliations:** 1School of Electronic and Electrical Engineering, Kyungpook National University, Daegu 41566, Korea; ehdnjs5169@knu.ac.kr (D.-W.K.); dan7620@knu.ac.kr (H.-J.K.); yongsz@knu.ac.kr (W.-Y.L.); kkd9506@knu.ac.kr (K.K.); sinhlee@knu.ac.kr (S.-H.L.); jhbae@ee.knu.ac.kr (J.-H.B.); imkang@ee.knu.ac.kr (I.-M.K.); 2School of Electronics Engineering, Kyungpook National University, Daegu 41566, Korea; 3School of Electronics and Information Engineering, Korea Aerospace University, Goyang 10540, Korea; kke@kau.ac.kr

**Keywords:** sol–gel, Y_2_O_3_, RRAM, surface roughness, local electric field, reliability

## Abstract

Sol–gel-processed Y_2_O_3_ films were used as active channel layers for resistive random access memory (RRAM) devices. The fabricated ITO/Y_2_O_3_/Ag RRAM devices exhibited the properties of conventional bipolar memory devices. A triethylamine stabilizer with a high vapor pressure and low surface tension was added to realize the local electric field area. During drying and high-temperature post-annealing processes, the large convective flow enhanced the surface elevation, and the increased –OH groups accelerated the hydrolysis reaction and aggregation. These phenomena afforded Y_2_O_3_ films with an uneven surface morphology and an increased surface roughness. The increased roughness of the Y_2_O_3_ films attributable to the triethylamine stabilizer enhanced the local electrical field, improved device reliability, and achieved successful repetition of the switching properties over an extended period.

## 1. Introduction

Resistive random access memory (RRAM) has been a promising core component for next-generation memory applications. Its simple metal-active material–metal structure can considerably increase the device density while overcoming Moore’s law with a low power consumption and a high device performance [1,2,3]. These advancements are particularly well suited to the requirements of neuromorphic computing systems, which require an enhanced device density and high performance to realize brain-like operations. ZrO_2_, TiO_x_, HfO_x_, and Y_2_O_3_ have been employed as active materials for RRAM devices [4,5,6,7,8,9,10]. Among these, Y_2_O_3_ shows a relatively high dielectric constant, large optical bandgap with thermal stability, and fast ion transport inside Y_2_O_3_. Hence, Y_2_O_3_ has been considered an alternative high-k insulator to conventional SiO_2_ insulators in complementary metal–oxide-semiconductor processes. Furthermore, Y_2_O_3_ has emerged as a promising active material for preparing RRAM devices [10,11,12]. Forming-free RRAM devices can be prepared using oxygen vacancy (O_v_)–rich Y_2_O_3_ [13,14]. The forming process requires an additional large voltage for the initial conductive filament (CF), occasionally resulting in electrode destruction. A synaptic memory cell was fabricated successfully using only a single active layer, i.e., Y_2_O_3_ [13]. Unlike other RRAM devices, Y_2_O_3_-based RRAM devices do not undergo device performance degradation owing to oxygen bubble formation [15], toxicity [16], thermal instability [16], and pulsed voltage stress [17]. In single RRAM devices, the sneak path issue is critical for achieving high-density arrays. To address this issue, the addition of a thin-film transistor, a 1-transistor–1-RRAM (1T–1R) structure, is suggested [18,19]. Recently, a Y_2_O_3_ layer was employed to solve the bias instability problem in thin-film transistors using a passivation layer [20]. Consider the case in which Y_2_O_3_ is simultaneously used as the active material and the passivation layer for RRAM devices and thin-film transistors, respectively. In this case, the processing time and cost of 1T–1R unit structures can be efficiently reduced in RRAM arrays. One of the main mechanisms for switching RRAM devices involves the formation and rupture of conductive paths between the top and bottom electrodes using a voltage bias to change the resistance between the two electrodes. The electrical field can affect the performance of RRAM devices. The resistive switching properties of RRAM devices are based on the electrical field-dependent charge injection and transport through the active channel layer [21,22]. In flat electrode–based RRAM devices, the applied field is distributed over the entire flat electrodes, leading to the formation of multiple random conductive paths. This makes it hard to control them satisfactorily, resulting in poor and inconsistent device performance because of the random nucleation, growth, and rupture of the conductive paths. This randomness also hinders repetitive switching cycles, such as endurance properties, thus affecting the nonvolatility of RRAM devices [23,24]. To address this issue, hourglass-shaped electrodes have been used instead of flat or nanocrystal-covered electrodes to facilitate local electric field biasing, in which the geometric advantages are exploited to improve the switching reliability by reducing the random formation and growth of conductive paths [25,26,27,28,29]. However, the formation of hourglass-shaped or nanocrystal-covered electrodes requires additional photolithographic or deposition processes; for instance, an additional patterning process is required for metal island deposition, and an etching process is required to obtain an hourglass shape.

The sol–gel process is an easy and simple approach for forming high-quality metal–oxide layers. Furthermore, liquid-phase precursors provide the ink required for spin-coating, dip-coating, and printing processes, which is beneficial for large-area application without the need for any conventional vacuum-based high-cost deposition tools [20,30,31]. Interestingly, the morphology of films can be easily controlled by changing the physical and chemical properties of the precursor, such as the pH, aging time, viscosity, stabilizer, drying conditions, and concentration. Thus, the wavy structure that triggers the generation of the local electrical field can be formed easily and simply without using additional and complex processes.

Herein, RRAM devices were fabricated using sol–gel-processed Y_2_O_3_ active layers. A triethylamine stabilizer with a high vapor pressure and low surface tension was added to realize the local electrical field area. During the annealing process, the large convective flow enhanced the surface elevation, and the increased –OH groups accelerated the hydrolysis reaction and aggregation, thus increasing the surface roughness. The increased roughness of the Y_2_O_3_ films owing to the addition of the triethylamine stabilizer afforded an enhanced local electrical field, improved device reliability, and the long and successful repetition of the switching properties.

## 2. Materials and Methods

Yttrium (III) nitrate tetrahydrate, 2-methoxyethanol (2-ME), ethanolamine (MEA), triethylamine (TEA), and ITO-coated glass were purchased from Sigma Aldrich, and acetone was purchased from DukSan (Seoul, Korea). Yttrium (III) nitrate tetrahydrate and 2-ME were used as the yttrium (Y) source and solvent, respectively. MEA and TEA were used as stabilizers. Solutions required in the sol–gel process were prepared under three conditions: without any stabilizer (REF), with MEA as the stabilizer, and with TEA as the stabilizer. The Y content in all the solutions was fixed at 0.3 M. The REF solution was prepared by mixing 10 mL of 2-ME and 0.3 M Y (III) nitrate tetrahydrate. To prepare the MEA and TEA solutions, the Y:MEA and Y:TEA molar ratios in the total solution volume of 10 mL were 1:1, respectively. All the solutions were sonicated for 1 h to afford clear solutions. The RRAM device used herein shows an ITO/Y₂O₃/Ag structure. Commercial ITO-coated glass was soaked in acetone and deionized water for 10 min each using a sonication process. To remove undesired organic dust, the ultraviolet/ozone process was performed for 1 h. The prepared Y_2_O_3_ precursor solution was coated on the ITO-coated glass substrate. To dry the solvent and retain the stabilizer, a soft drying process was performed on a hot plate at 150 °C for 10 min. To prepare Y_2_O_3_ films, the post-annealing process was conducted at 500 °C for 2 h. After letting the films cool gradually, the top Ag electrode with a thickness of 150 nm was deposited on the films via thermal deposition. The patterning of the top Ag electrode was performed using a shadow metal mask. The deposited Ag pattern comprised a square with a side of 30 μm. The structural properties of the films were analyzed using grazing incidence X-ray diffraction (GIXRD, X’pert Pro, Malvern PANalytical, Malvern, UK; Cu Kα, λ = 1.54 Å, incident angle = 0.3°). The surface morphology and thickness of the films were investigated using scanning probe microscopy (SPM, Park NX20, Park Systems, Suwon, Korea) and field emission scanning electron microscopy (FE-SEM, Hitachi 8230, Hitachi, Tokyo, Japan). X-ray photoelectron spectroscopy (XPS; Nexsa, ThermoFisher, Massachusetts, U.S.A) was used to examine the chemical properties of the oxide thin films. The electrical properties―such as current–voltage (I–V) curves, endurance, and retention―were measured using a semiconductor parameter analyzer (Keithley 2636B, Keithley Instruments, Cleveland, OH, USA) and a probe station (MST T-4000A, Hwaseong, Korea).

## 3. Results and Discussion

Figure 1 and Figure 2 show the atomic force microscopic and SEM images of the films prepared using different stabilizers after the high-temperature annealing process. All the films were continuous; however, the surface morphologies of the films differed based on the stabilizer. The films obtained using stabilizer-added precursor solutions exhibited large aggregated and wavy structures. The TEA stabilizer afforded films with the roughest surface and most aggregated morphology. The formation of the aggregated and wavy structure can be attributed to stress relaxation, frequently observed in microscale and nanoscale film structures.

The film uniformity was affected by the physical properties of the two solvents used. Here, a co-solvent system was used to form Y_2_O_3_ films (Table 1). One solvent with a high vapor pressure and low surface tension was mixed with another solvent that showed a low vapor pressure and high surface tension. The solvent with a high vapor pressure was preferentially evaporated from the film surface. The remaining solvent induced a net increase in surface tension. The increased surface tension triggered a large convective flow and increased the surface elevation [32,33,34].

Furthermore, in the high-temperature post-annealing process, above the boiling point of the solvents, the partially-thickened portion of the film exhibited enhanced strain owing to different thermal expansions. This strain was intensified at the partially thickened part of Y_2_O_3_. Thus, the Y_2_O_3_ films formed using the TEA co-solvent system showed the most aggregated and wavy morphology. The solution pH can also affect the film morphology. The pH levels of the REF-, MEA-, and TEA-added precursor solutions were 4.8, 6.2, and 6.4, respectively. Solutions with a low pH level contain more H^+^ than those with a high pH level. This induces the formation of hydronium ions (H_3_O^+^), hindering both the hydrolysis and condensation processes. Alternatively, a solution with a high pH level contains many OH– ions, affording an uneven surface morphology because of the accelerated hydrolysis reaction and aggregation [35].

Figure 3 shows the GIXRD spectra of the annealed films. At the bottom of the GIXRD spectra, cubic Y₂O₃ were observed with peaks indicated by purple bars; particularly for the annealed films, peak values (2*θ*) of Y_2_O_3_ were detected. The annealed films showed peaks, and each film achieved the same peak degree value, indicating that the annealed films were crystallized and oriented along the same grain-developed direction. The cubic structure was observed throughout the Y₂O₃ films (JCPDS 41-1105). In the GIXRD pattern, the cubic structure of the Y₂O₃ film showed peaks at 2*θ* = 29.150°, 33.784°, 48.541°, and 57.622°, corresponding to the (2 2 2), (4 0 0), (4 4 0), and (6 2 2) planes, respectively, ensuring the polycrystalline structure of the Y₂O₃ layer. The crystalline sizes of the films can be estimated using the following Scherrer equation:(1)D=(0.9λ)/(βcosθ)
where *D, λ, β*, and *θ* indicate the crystalline size, X-ray wavelength (1.54 Å), full width at half the maximum of each diffraction peak, and peak position, respectively. The crystalline sizes calculated for the (2 2 2) plane were 11.4 nm (REF), 10.7 nm (MEA), and 9.6 nm (TEA).

Figure 4 shows the chemical composition of the films analyzed using XPS. The XPS peaks were examined based on the Gaussian distribution. Figure 4a–c shows the O1s spectra, indicating three peaks that represented the oxygen ions combined with metal cations (O_L_), surface O_V_, and CO binding groups, with binding energies of 529.0, 530.7, and 532.1 eV, respectively. For the C–O binding groups, the binding portion increased from 0% for REF to 39% for MEA, and 57% for TEA after the addition of stabilizers. In Figure 4a, the Y3d peak comprised Y3d_5/2_ and Y3d_3/2_ components, with binding energies of 155.9 and 157.8 eV, respectively. These findings confirm the successful formation of the Y₂O₃ films. Moreover, no peak related to –Y–OH (~160 eV) was detected in the aforementioned spectra.

Figure 5 shows the I–V curve of the RRAM devices based on the Y_2_O_3_ films formed using different stabilizers. The I–V curve was analyzed under DC voltage, sweeping along the sequence of 4 → 1 → 2 → 3 in Figure 5. The DC voltage sweeping ranges were −16.0–+5.0 V and +5.0–−16.0 V, with an increment and decrement of 0.05 V, respectively. The bottom electrode was grounded, and voltage was applied at the top electrode (Ag). All fabricated RRAM devices showed the characteristics of conventional bipolar RRAM switching. Several Y_2_O_3_-based RRAM devices have shown that the resistive switching memory behavior can be attributed to the formation and destruction of an O_v_–based conductive path. ITO/Y_2_O_3_/Au RRAM devices do not show these characteristics of RRAM devices [14], indicating that the formation and rupture of the metallic conductive path, originating from the oxidation and reduction processes from Ag electrodes, are the main mechanisms of ITO/Y_2_O_3_/Ag RRAM devices. The fabricated devices showed a high-resistance state (HRS). When the positive voltage was biased toward the top electrodes, Ag atoms were oxidized, forming Ag ions (Ag → Ag^+^ + e^−^). Ag^+^ ions drifted when the electric field reached the bottom electrode, and Ag^+^ received an electron from the cathode, reducing to Ag atoms. Finally, Ag^+^ ions became metallic CFs, consisting of Ag atoms. These CFs reached the electrodes, transitioning to the low-resistance state (LRS) at a SET voltage. The current in the RRAM device at arrow 3 in Figure 5a decreased drastically because of the rupture of the metal CFs. When the LRS was converted to the HRS, the Ag CFs underwent thermal-assisted oxidation, inducing the transition of Ag atoms to Ag^+^ ions. The oxidized Ag^+^ ions drifted to the top electrode owing to the electric field based on the negative voltages applied on the top electrode. Finally, the metallic CFs ruptured, decreasing the current; then, the HRS was returned to a RESET voltage. The HRS value was larger in the order of TEA, MEA, and REF. The HRS/LRS ratio of the TEA, MEA, and REF films were approximately 7.97 × 10^4^, 3.38 × 10^5^, and 2.32 × 10^5^, respectively. The RESET voltage was higher than the SET voltage. The voltage ramp rate (V/s) critically influences the RESET voltage [36,37,38]. A fast ramp rate affords a high RESET voltage in the system.

Based on the GIXRD data, the REF sample showed the largest crystalline size. The carrier mobility, related to the leakage current of the insulator, increases with an increase in the grain size, and is inversely proportional to the number of grain boundaries. The enhanced carrier scattering caused by the grain boundary decreases the leakage current. Based on the XPS data, the MEA and TEA samples showed lower O_v_ proportions compared to REF samples. O_v_ is generally recognized as a defect in oxides, acting as a trap to enable trap-assisted tunneling or hopping mechanisms. The suppression of O_V_ suppresses the current flow, which decreases the leakage current by eliminating the defects in the oxide films. These phenomena decrease the leakage current and increase the HRS. Figure 5b shows the average values of the SET and RESET voltages. The devices fabricated using the TEA-added precursor solution showed the lowest SET and RESET voltages. The comprehensive charge concentration, and efficient charge injection and transport afforded a low-voltage operation through the enhanced local electrical field [28,29]. The cumulative probabilities of the RESET and SET voltages are plotted in Figure 5c,d, respectively. Compared with the REF device, the MEA and TEA devices showed a statistically improved uniformity in the SET and RESET voltages.

The performance of the nonvolatile RRAM devices was estimated by analyzing their retention and endurance properties (Figure 6). The LRS and HRS were measured at V_read_ = +0.1 V after the program (+4.0 V for 500 ms) and erase (−17.0 V for 500 ms) operations. The LRS and HRS lasted for ~10^4^ s without any considerable degradation. Moreover, the RRAM devices based on the Y_2_O_3_ films obtained from the stabilizer-added precursor solutions showed enhanced endurance properties.

The rough Y_2_O_3_ films successfully enhanced the electric field at the summit of the films and resulted in charge injection and transport, thus forming a location-controlled conductive path. When a potential was applied to the wavy structure, the charges concentrated at the sharp part rather than the plane part. Based on a simple electrostatic model, the surface charge density is expressed as:(2)σ=q/4πr,
where *q* indicates the charge quantity and *r* represents the radius of the structure. The charge density can also affect the generated electric field:(3)$$\nabla \cdot E=\varrho /\varepsilon_{0},$$
where *ρ* denotes the charge density and *ε_0_* represents the permittivity of dielectric materials. These equations indicate that additional charges are concentrated at the sharp structure, enhancing the electric field at the summit of the rough films. The estimated radii of the cylindrical structures based on MEA and TEA were ~120 and 80 nm, respectively. The TEA films could generate a more pronounced local electrical field than the plane REF films.

The rougher Y_2_O_3_ films between two electrodes resulted in improved endurance properties, whereas the smoother Y_2_O_3_ film between the two electrodes in the REF device could generate and inject cations randomly in the entire electrode area, yielding randomly-formed multiple CFs between the two electrodes. By concentrating the biased field in a specific area, the rough Y_2_O_3_ films between the two electrodes could suppress the formation of random multiple CFs. The increased number of CFs hinders the trigger of the SET and RESET operations, because all formed CFs should be simultaneously controlled using external bias voltages to confirm the constant SET and RESET voltages. If a single CF cannot be controlled using the external bias voltages, the SET and RESET operations fail and increased SET or RESET voltage values are required, leading to poor operation uniformity. Further, during the endurance test, uncontrolled CFs decreased the endurance cycle. Thus, the RRAM device based on the roughest Y_2_O_3_ films (TEA) showed improved SET and RESET uniformity and enhanced switching reliability, as well as endurance.

## 4. Conclusions

Sol–gel-processed Y_2_O_3_ films were used as active channel layers for RRAM devices. The fabricated ITO/Y_2_O_3_/Ag RRAMs show conventional bipolar memory properties. A high-vapor-pressure and low-surface-tension triethylamine stabilizer was introduced to obtain the local electrical field area. During the drying and high-temperature post-annealing processes, the large convective flow enhanced the surface elevation, and the increased –OH groups accelerated the hydrolysis reaction and aggregation, thereby increasing the roughness and uneven morphology of the Y_2_O_3_ film surfaces. The increased roughness of the Y_2_O_3_ films caused by adding the triethylamine stabilizer yielded an enhanced local electrical field, improved device reliability, and the successful and long repetition of the switching properties.

## Figures and Tables

**Figure 1 materials-15-01943-f001:**
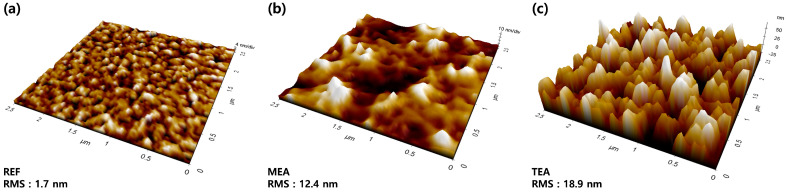
SPM images of the films prepared using different stabilizers after the high-temperature annealing process: (**a**) REF, (**b**) MEA, and (**c**) TEA.

**Figure 2 materials-15-01943-f002:**
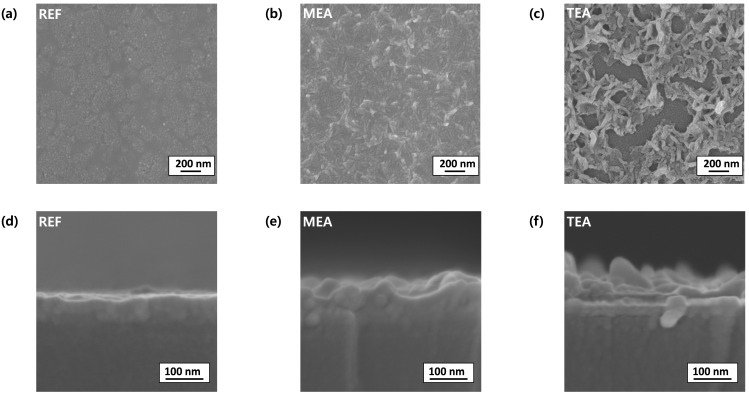
Top-view SEM images ((**a**) REF, (**b**) MEA, and (**c**) TEA), and side-view SEM images ((**d**) REF, (**e**) MEA, and (**f**) TEA) of the films prepared using different stabilizers.

**Figure 3 materials-15-01943-f003:**
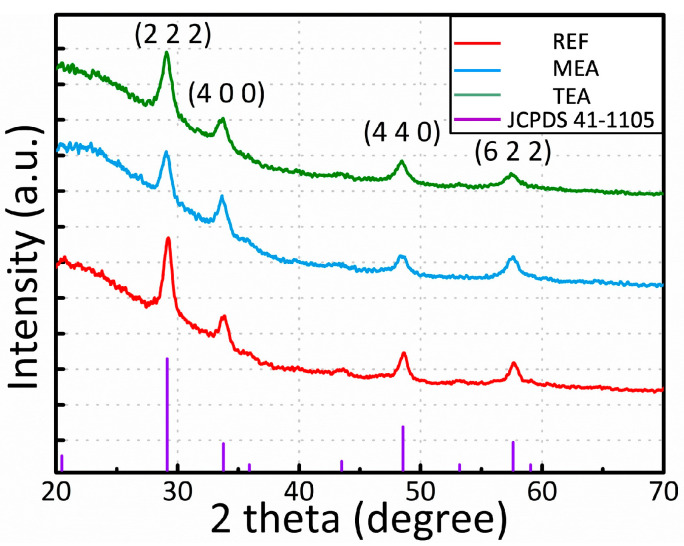
GIXRD spectra of the sol–gel-processed Y_2_O_3_ films with different stabilizers.

**Figure 4 materials-15-01943-f004:**
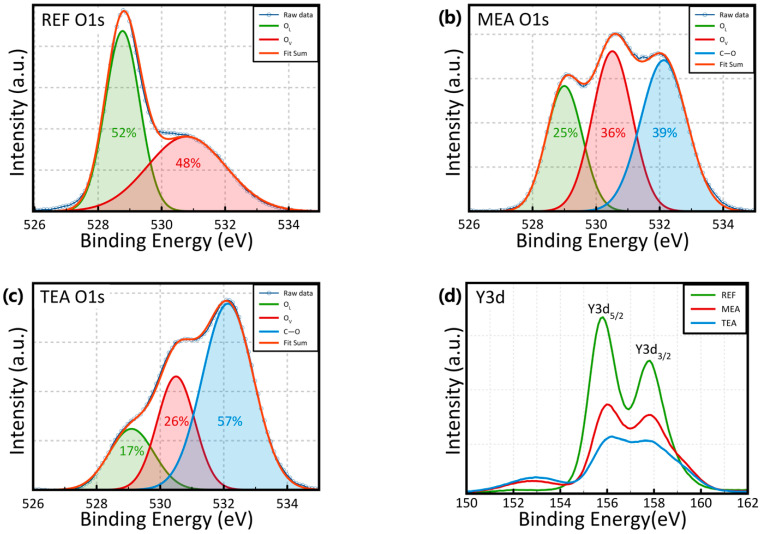
XPS patterns of (**a**–**c**) O 1s and (**d**) Y3d spectra of the sol–gel-processed Y_2_O_3_ films with different stabilizers.

**Figure 5 materials-15-01943-f005:**
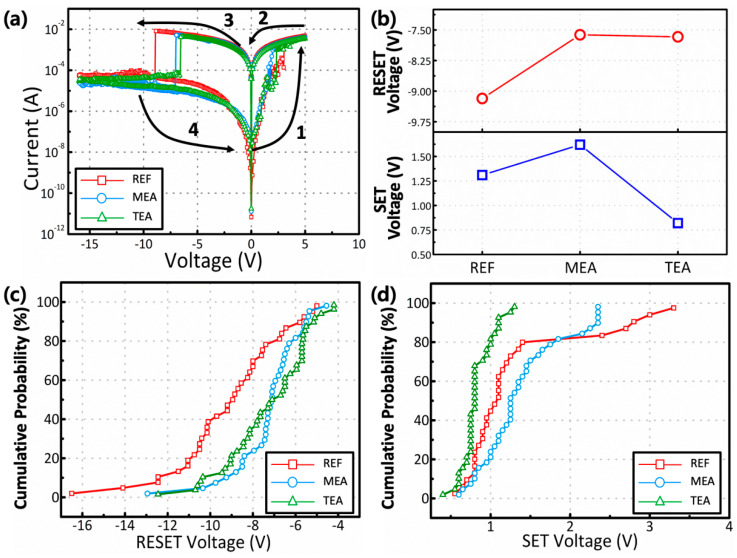
(**a**) Representative I-V curves of the fabricated Y_2_O_3_ RRAM devices. The arrows and numbers indicate the voltage bias directions. (**b**) average values of the RESET (red empty circle) and SET (blue empty square) voltages, and (**c**) cumulative probability graph of the RESET voltage and (**d**) SET voltage for devices fabricated using different stabilizers.

**Figure 6 materials-15-01943-f006:**
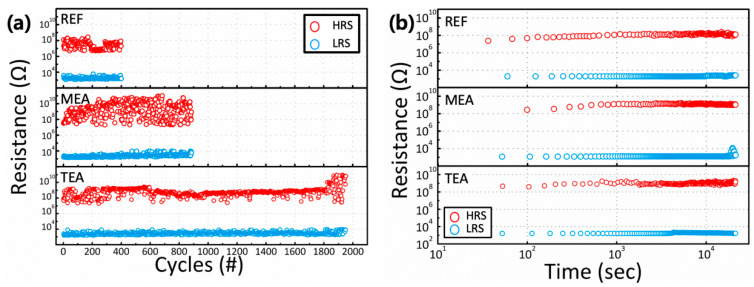
Nonvolatile properties of Y_2_O_3_ RRAM devices prepared using different stabilizers: (**a**) Representative endurance and (**b**) retention time.

**Table 1 materials-15-01943-t001:** Surface tension, boiling point, and vapor pressure of the solvents.

Name (Abbreviation)	Surface Tension (N/m) at 25 °C	Boiling Point (°C)	Vapor Pressure (mm Hg) at 25 °C
2-methoxyethanol (2-ME)	42.8	124.0	9.5
Ethanolamine (MEA)	48.3	170.0	0.4
Triethylamine (TEA)	22.02	88.8	57.07

## Data Availability

Data are available in a publicly accessible repository.

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
