# Peer review of "Enhanced Switching Reliability of Sol–Gel-Processed Y2O3 RRAM Devices Based on Y2O3 Surface Roughness-Induced Local Electric Field"

_materials, 2022, doi:10.3390/ma15051943_

Round 1

Reviewer 1 Report

While the paper addresses a topic of sufficient interest and importance, it suffers from several shortcomings.

1) The authors debate the possibility of locally high electric field in their structures as an important factor determining the device operations. However, the question of why and how the field can be beneficial is not addressed. Some references dealing  with the field issues would be appropriate here. For example, Niraula and Karpov, J. Appl. Phys. 124, 174502 (2018);  Karpov et al,  Phys. Rev. Applied, 8, 024028 (2017).

2) The authors should at least try a more quantitative (even rough) estimate of what is the field enhancement they achieved.

3) They should at least qualitatively discuss how the strong local field advent can affect SET and RESET transitions and explain whether these agree with their results.

In view of he above concerns, I think that the paper should be significantly revised before considered for publication. Noticeable language issues should be taken care of as well.

Reviewer 2 Report

The authors reported enhanced switching reliability in ITO/Y2O3/Ag RRAMs. The resistive switching performances were investigated and discussed. The reported results are interesting and this study is of relevance to the resistive switching devices. However, the mechanism is unclear and the present version of the manuscript needs moderate revision. The authors should address the follow questions before consider for publication in the journal.
Comments:
1. Some relevant papers in resistive switching field have been published in recent years. Some important and latest research results in this field should be mentioned and cited in the section of introduction instead of outdated or earlier papers so that we can offer a solid background and progress to the readers regarding the current state of knowledge on this topic. Therefore, I strongly require the authors to revise this part.
2. The Y2O3-based resistive switching devices have potential application in RRAM. The authors should give more comments in the introduction section to present the innovativeness and significance in their works.
3. In the Results and Discussion section, the fluctuation of RESET/SET voltage characteristics in devices is large in Fig. 5(c-d), so it needs lot of improvement.

4. What about the switching speeds characteristics in devices? This characteristic of the thin films are crutial to evaluate RS performance. The authors need to add this characteristic and give some comments.
5. The authors explain the RS mechanism, whereas the mechanism is not clear. The authors need to clear explain RS mechanism by providing some evidences to justify in their revision, such as observation of conducting filament growth by TEM method. Moreover, the authors need to cite more relative papers of their manuscript and discuss the RS mechanism.
6. Finally, there are some grammar mistakes and confusing sentences in the manuscript. The authors need to be corrected.

Round 2

Reviewer 2 Report

The authors have made a suitable revision according to the comments by referees. The manuscript is acceptable for publication.